# Asymptomatic Malaria Infection and Hidden Parasitic Burden in Gabonese Schoolchildren: Unveiling Silent Co-Infections in Rural and Urban Settings

**DOI:** 10.3390/tropicalmed10010011

**Published:** 2024-12-31

**Authors:** Patrice Makouloutou-Nzassi, Lady Charlene Kouna, Chérone Nancy Mbani Mpega Ntigui, Neil Michel Longo-Pendy, Judy Armel Bourobou Bourobou, Felicien Bangueboussa, Nick Chenis Atiga, Jean Bernard Lekana-Douki, Larson Boundenga, Sandrine Lydie Oyegue-Liabagui

**Affiliations:** 1Département de Biologie et Ecologie Animale, Institut de Recherche en Ecologie Tropicale (IRET/CENAREST), Libreville BP 13354, Gabon; 2Unité de Recherche en Ecologie de la Santé, Centre Interdisciplinaire de Recherches Médicales de Franceville, Franceville BP 769, Gabon; longo2michel@gmail.com (N.M.L.-P.); bangueboussafelicien040@gmail.com (F.B.); boundenga@gmail.com (L.B.); 3Unit of Evolution, Epidemiology and Parasite Resistance (UNEEREP), Franceville Interdisciplinary Center for Medical Research (CIRMF), Franceville BP 769, Gabon; charleneklc@gmail.com (L.C.K.); mpega_mb2@yahoo.fr (C.N.M.M.N.); atiganick@yahoo.fr (N.C.A.); lyds_ass@yahoo.fr (S.L.O.-L.); 4Institut de Recherches Agronomique et Forestière (IRAF/CENAREST), Libreville BP 2246, Gabon; judiarmelbourobou@gmail.com; 5Department of Parasitology-Mycology, University of Health Sciences (USS), Libreville BP 4009, Gabon; lekana_jb@yahoo.fr; 6Central African Regional Doctoral School in Tropical Infectiology (ECODRAC), Franceville BP 876, Gabon; 7Department of Anthropology, Durham University, South Road, Durham DH1 3LE, UK; 8Department of Biology, Faculty of Science, Masuku University of Science and Technology (USTM), Franceville BP 876, Gabon

**Keywords:** anemia, children, co-infections, Gabon, Lastourville, malaria infection

## Abstract

This study aimed to determine the prevalence of co-infection with malaria and intestinal parasites and assess its association with anemia in school-aged children from rural and urban settlements in Gabon. This cross-sectional study involved afebrile school children recruited at schools between May and June 2021. Blood and stool samples were collected from participants whose parents or legal guardians provided informed consent to participate in the study. Hemoglobin concentration (Hb) was measured using a HemoCue photometer (HemoCue 201, HemoCue, Angelholm, Sweden). Giemsa-stained blood films were examined to detect malaria parasites and any filarial infections, while the merthiolate-iodine concentration (MIC) method was used to identify intestinal parasitic infections (IPI). A total of four hundred and seventy (470) school-aged children were successfully enrolled in this study. The observed prevalence values were as follows: malaria infection at 69.6%, IPIs at 19.1%, filaria at 5.1%, *Schistosoma* infection at 15.0%, and anemia at 29.0%. Co-infections of malaria with IPIs, filaria, and *Schistosoma* were present in 12.3%, 4.7%, and 6.6% of the children, respectively. Malaria and filaria infections were associated with residing in Lastourville city (*p* < 0.05) and were also correlated with age (*p* < 0.05), whereas IPIs were associated with male gender and living in the city of Lastourville. Anemia was linked to malaria infection (*p* < 0.05) and was more prevalent among children living in rural areas. The findings of this study indicate that malaria, IPIs, and *Schistosoma* infections continue to pose a significant public health problem in the study area, even though only malaria infection appeared to be associated with anemia. Nevertheless, these results highlight the need for implementing control measures to reduce the prevalence of malaria, IPIs, filaria, and *Schistosoma*, particularly in Lastourville.

## 1. Introduction

*Plasmodium* and intestinal parasitic infections are significant public health issues in tropical and subtropical settings, particularly in poor communities with inadequate sanitation and hygiene. Given that the distribution of these parasitic diseases spatially overlaps, co-infections in the same individual are common and can result in severe morbimortality. Indeed, *Plasmodium* infection (with malaria) is a leading cause of death, especially among children. In 2020, the World Health Organization (WHO) recorded approximately 241 million clinical cases and 627,000 deaths from malaria worldwide. Sub-Saharan Africa accounts for 95% of malaria cases and 96% of malaria deaths, with children accounting for 80% of all malaria deaths [1]. Whereas Intestinal parasitic infections (IPI), caused either by soil-transmitted helminths (STH), protozoa, or both, are responsible for 450 to 840 million cases worldwide, and the majority are reported in developing countries [2]. Although malaria affects human life negatively, *Plasmodium falciparum* infections manifest through heterogeneous outcomes ranging from asymptomatic infection to severe disease, which may rely on the parasite threshold [3,4] as well as the nutritional status of the host [5]. The WHO defined asymptomatic malaria infection as the presence of asexual parasites in the blood without symptoms of illness [6]. For Lindblade et al., asymptomatic malaria is the existence of malarial parasitemia of any density in blood without any symptoms in individuals who have not received recent anti-malarial treatment in each population [7]. Whereas Bousema et al. argue that this definition should include early detection of rising parasitemia or any density of parasitized red blood cells (RBCs) that is not enough to trigger a fever response [3]. In Gabon, where malaria transmission is perennial, studies have investigated the distribution of asymptomatic malaria in the country. So far, most of the studies conducted in Gabon were cross-sectional studies [8,9,10,11,12] and involved children [10,12], or adults [9,11]. They were mostly carried out in rural settings [11,12]. Nevertheless, one longitudinal survey was conducted on school aged children (SAC) in rural settlements [13]. Besides studies reporting the distribution of asymptomatic malaria in a given population, some surveys have explored the related immune response [8,14], the genetic diversity of *Plasmodium* in infected individuals with or without sickle cell disease [12], the human genetic polymorphisms and the prevalence and profile of asymptomatic malaria [10], the profile of 10 cytokines in asymptomatic malaria children living in different settings [15]. However, there are little data regarding the association between asymptomatic malaria and IPIs. However, a study that investigated the effect of schistosomiasis and STH on the prevalence and incidence of *P. falciparum* infection highlighted that STH enhances the risk for *Plasmodium* infection in schistosomiasis-positive children and, when infected, that schistosomiasis enhances susceptibility to developing malaria in young children but not in older children [13]. Despite previous studies, our understanding of the epidemiology of this co-infection in Gabon remains limited. There is particularly a lack of data on the co-infection of *Plasmodium* and IPIs in urban areas like Franceville and rural areas like Lastourville. This study aims to assess the prevalence and determinants of asymptomatic malaria infection and IPIs in schoolchildren in different settlements. The findings of this study could provide valuable insights to Gabonese health authorities for improving current control strategies.

## 2. Materials and Methods

### 2.1. Study Sites and Participants

The study was carried out in two regions of Gabon situated in two provinces of the country: Franceville (province of Haut-Ogooué) (1°37′15″ S and 13°34′58″ E) and Lastourville (province of Ogooué-Lolo) (0°49′ S, 12°42′ E). Franceville is the third-largest city in Gabon, whereas Lastourville is a rural agglomeration of several villages. Gabon has an equatorial climate which consists of a short dry season (from 15 June to 15 September) and a long rainy season (from 15 September to 15 June). The study involved primary-aged children of both genders living in both regions whose parents or legal guardians consented to their participation in the study. Children’s involvement was voluntary, and only those who had lived at the study sites for at least 3 months were included. Before the data were collected, the research team visited the study sites to educate the local authorities and residents on the importance, benefits, and protocols of the research. This study was a cross-sectional study of a cohort of school-aged children between the ages of 3 and 17 enrolled in randomly selected elementary schools at both sites. Only children with signed informed consent from a parent or legal guardian were included. The participants underwent interviews and medical examinations before providing blood, feces, and urine samples. Blood was collected in ethylenediaminetetraacetic acid (EDTA) tubes, and feces were collected in clean, well-labeled stool vials at both study sites. Urine samples were collected only from participants in Lastourville using plastic screw-cap vials. Blood samples were analysed for malaria and filariasis, feces for the presence of STH eggs, and urine was used for the microscopic detection of *Schistosoma haematobium*. 

### 2.2. Blood Sample Examination

#### 2.2.1. Thick Blood Films

For microscopic detection, thick blood smears (TBS) were prepared and used as described elsewhere [16]. After staining with 20% Giemsa solution for 20 min, the slides were examined under a 100× oil-immersion objective. Parasitemia was calculated for all positive TBS, as the number of parasites per microlitre of blood. If no parasites were found after examining 100 oil immersion fields, the slides were considered negative.

#### 2.2.2. Detection of Microfilaria 

The detection of microfilaria was carried out using the Sang–Petithory leucoconcentration technique [17]. All microfilariae found on the slide were identified. Due to its higher sensitivity, this technique is indicated when the parasite density is low. Leucoconcentration was applied to all participants who provided blood samples.

### 2.3. Fecal Examination by Merthiolate-Iodine Concentration (MIC)

Fecal samples were analyzed using the merthiolate-iodine concentration (MIC) method, described by Sapero and Lawless [18], but slightly modified. Briefly, 20 mL of distilled water was added to the plastic screw cap vials containing around 1 g of each sample and homogenized using an applicator stick. In total, 5 mL of this preparation were collected and transferred into a conic tube, then centrifuged for 4 min at 3000× rpm. The supernatant was discarded by rapidly inverting the tube, and 500 µL of distilled water was added, then mixed well. A drop was collected and placed on a glass slide and blended with a drop of MIC solution. Covered, the entire preparation under the cover slide was examined after 1 min for eggs, cysts, and larvae using a 10× and 40× lens of a light microscope. Each slide was examined in duplicate by two experienced technicians. Once only one parasite was found, the sample was considered positive. 

### 2.4. Urine Screening for Schistosome Eggs

Urine samples were reviewed using the centrifugation method previously described by Deriben et al. [19]. Haematuria was determined either visually (general haematuria) or microscopically (microhaematuria). As stated above, microscopic detection of *S. haematobium* eggs was performed on urine samples obtained from participants in Lastourville only.

### 2.5. Measuring the Haemoglobin Concentration 

Hemoglobin (Hb) concentration was measured using a HemoCueH photometer (HemoCue 201, HemoCue, Angelholm, Sweden). Anemia was defined as a Hb level of ˂11 g/dL and further classified according to WHO anemia thresholds: severe anemia: Hb < 7 g/dL, moderate anemia: Hb 7–9.9 g/dL, mild anemia: Hb 10–10.9 g/dL [20]. 

### 2.6. Definitions and Endpoints

Asymptomatic malaria parasitemia was defined as the presence of *Plasmodium* in the blood by microscopic examination, with an axillary temperature of ˂37.5 °C and no record of fever in the past 2 weeks.Parasitemia was categorized as low (˂1000 parasites/µL of blood), moderate (1000–4999 parasites/µL blood), and high (≥5000 parasites/µL blood) [21].

### 2.7. Statistical Analysis

Management and tabulation of raw data were carried out using Microsoft Excel (Microsoft Inc., Redmond, WA, USA) version 2016. All statistical analyses were performed using R software R.4.3.2 (R Foundation for Statistical Computing, Vienna, Austria). Variables were expressed as proportions for categorical variables or as medians (with range)/means (standard deviations [SD]) for continuous variables. Frequencies (%) of participants’ sociodemographic data and the presence of *Plasmodium*, intestinal parasites, *Loa loa*, *M. perstans*, *Schistosoma haematobium* monoparasitism, and polyparasitism (including *Plasmodium* and co-infections) were determined. Chi-square or Fisher’s exact tests were used to assess differences in malaria infection, IPIs, filariasis, and *Schistosoma* frequencies, as well as to compare risk factors between and within age, sex, weight, body temperature, and location groups. The strength of association between each risk factor and the occurrence of *Plasmodium* infection, intestinal parasites, *Loa loa*, *M. perstans*, and *Schistosoma haematobium* was calculated using logistic regression models. The level of significance was set at *p* < 0.05.

## 3. Results

### 3.1. Study Population

The study included 470 school-aged children (SAC) from Franceville (urban) and Lastourville (rural). Their characteristics are summarized in Table 1, with 249 (53%) from urban areas and 221 (47%) from rural areas. There were 247 females (52.6%) and 222 males (47.4%), yielding a sex ratio of 0.9. The participants had a mean age of 10.04 (±3.2), with rural children being older on average than urban children (*p* = 0.03). Hematological parameters (hemoglobin, white blood cells, red blood cells) significantly differed between rural and urban areas (*p* < 0.0001), as did platelet counts (*p* < 0.001).

### 3.2. Pattern of Infection Diversity and Prevalence 

A total of 327 children tested positive for malaria parasites, resulting in a prevalence of 69.6% (95% CI: 65.4–73.7) (Table 2). The prevalence was significantly higher in rural areas (208/221 [94.1%]) compared to urban areas (119/249 [47.8%]) (*p* < 2.2× 10^−16^). Males had a *Plasmodium* infection rate of 71.2%, while the rate for females was 68.0% (*p* = 0.5218), regardless of location. Children aged 11 to 17 exhibited higher malarial parasitemia (162/211 [76.7%]) than those under eleven (163/250 [65.2%]) (*p* = 0.0089). Prevalence levels of low, moderate, and high parasitemia in the study population were 92.7% (303/327), 4.3% (14/327), and 3% (10/327), respectively. Overall parasitemia was also higher in rural areas (208/324 [64.2%]) than in urban areas (115/324 [35.5%]) (*p* = 4.893 × 10^−13^), regardless of parasitemia level. The parasite density varied from 3 to 41,450, with a mean ± SD of 1021.803 ± 3537.998.

Of the 470 participants, 402 children submitted stool samples, with 77 testing positive for intestinal parasites, yielding a prevalence of 19.2% (95% CI: 15.3–23.0). Infections were more prevalent in urban (24.7%) than in rural areas (13.3%) (*p* = 0.0055). The infection rate was higher in males (24.5%) compared to females (14.4%) (*p* = 0.0145). Children aged 11 to 17 had a higher prevalence (20.3%) than those under 11 (18.2%) (*p* = 0.680). *Ascaris lumbricoides* was the most common intestinal parasite (42.8%), followed by *Entamoeba coli* (41.5%), hookworms (16.8%), and whipworm (*Trichuris trichiura*) (10.3%). *Endolimax nana*, *Taenia saginata*, *Giardia* spp., and *Enteromonas hominis* were the least prevalent, at 3.8%, 2.5%, 2.5%, and 1.2%, respectively. 

A total of 24 children (5.1%) tested positive for microfilaria infection, with 3.4% for *Mansonella perstans* and 1.7% for *Loa loa*. Filarial infections were more prevalent in rural areas (10.1%) compared to urban areas (0.8%) (*p* = 1.522 × 10^−5^). The infection rates were 5.7% in females and 4.5% in males (*p* = 0.7273), regardless of location. Taking into account the age groups, prevalence was 2.4% for children aged 3 to 10 and 8.7% for those over 10 (*p* = 0.0055). No concomitant infections with *Loa loa* and *M. perstans* were observed in the study population.

Only children from Lastourville were screened for urinary schistosomiasis. Among the 212 participants, 15.1% (N = 32) tested positive based on egg observation via microscopy. The prevalence was 12.8% in females and 17.9% in males (*p* = 0.4046). Notably, 31 of the 32 schistosomiasis-positive children (96.8%) were from a location called “Gare de Setrag”.

### 3.3. Plasmodium, IPIs, Microfilariae, S. haematobium, and Co-Infection Pattern 

Globally, the prevalence of asymptomatic malaria co-infections with intestinal parasitic infections (IPIs) was 17.4% (57/327). Co-infections of malaria parasites with microfilaria reached 6.7% (22/327), while co-infections of malaria parasites and *Schistosoma haematobium* were at 14.9% (31/208).

Of the 57 children co-infected with asymptomatic malaria and intestinal parasitic infections (IPI), the most common combination was *Plasmodium* with *A. lumbricoides* (50.9%, or 29/57), followed by *Plasmodium* with *Entamoeba coli* (38.6%, or 22/57) and *Plasmodium* with *Ancylostoma* sp. (15.8%, or 9/57). *Plasmodium* co-infected with *Trichuris trichiura* occurred in 12.3% of cases (7/57), while rare co-infections with *Endolimax nana* were noted in only 3.5% of cases. Additionally, *Plasmodium* was found alongside *Giardia* sp. and *Taenia saginata* in single fecal samples. Instances of double parasitism included *Plasmodium* with *A. lumbricoides* (24 cases), *Plasmodium* with *Entamoeba coli* (15 cases), and less frequently, with *Ancylostoma* sp. (5 cases), *Trichuris trichiura* (2 cases), and *Endolimax nana* (1 case). There were also a few cases of triple parasitism: one child had *Plasmodium*., *Ancylostoma* sp., and *Giardia* sp., while another had *Plasmodium*, *Entamoeba coli*, *Ancylostoma* sp., and *Taenia saginata*, all identified in stool samples from the urban area.

Notably, all children with microfilaria infections were simultaneously infected with malaria parasites, but no children harbored concurrent infections of both *Loa loa* and *Mansonella perstans* alongside *Plasmodium*.

### 3.4. Risk Factors Associated with Malaria, IPI, Filaria 

Univariate analysis revealed that the risk of malaria infection was significantly associated with age (OR = 1.124, 95% CI: 1.054–1.200, *p* < 0.001). Furthermore, the risk of malaria infection was markedly higher in children living in rural areas (OR = 17.627, 95% CI: 9.882–33.996). Interestingly, gaining one kilogram in weight was linked with an increased risk of malaria infection, although this association was marginally significant (*p* = 0.037). The multivariate logistic regression model showed that both age and place of residence (City) were associated with increased risk of malaria infection (Table 3).

A logistic regression model (detailed in Table 4) demonstrated that the factors associated with intestinal parasitic infections in the univariate model were similarly significant in the multivariate model. Male gender and residing in the city of Lastourville (LTV) emerge as key factors linked to a heightened risk of intestinal parasitic infection. 

For filarial infection, a logistic regression model with gender, age, weight, temperature and city as independent variables, indicated that age and residing in LTV (*p* < 0.001) are significant risk factors for filarial infection. The odds of carrying filarial infection are presented in Table 5. 

There were no significant associations between asymptomatic malaria and intestinal parasitic infection or filaria (*p* > 0.05, Table 6). However, a significant relationship was found between asymptomatic malaria and *Schistosoma* infection (*p* < 0.05, Table 6).

### 3.5. Prevalence and Risk Factors for Anemia 

In this study, the overall prevalence of anemia was found to be 29.0% (110/379, 95%CI: 24.43–33.57), with no difference between sexes (*p* = 0.59). The prevalence rates of mild and moderate anemia were 62.7% (69), and 36.4% (40), respectively. One child had severe anemia (0.9%). The anemia rates were not influenced by the age category (*p* = 0.0246). Moreover, the prevalence of anemia was higher in rural areas (41.2%) compared to urban areas (22.5%). Among children diagnosed with asymptomatic malaria, a noteworthy 93 (38.1%) were anemic. A significant association was established between anemia and malaria infection (see Table 7).

### 3.6. Different Parasites and Polyparasitism

Polyparasitism, i.e., infection with more than one parasite, was detected in this study. Infection with multiple parasites (blood, stool and/or urine parasites) in the study population was 24.4% (99/405). Twelve different parasites were identified, among which seven helminths (*Ascaris lumbricoides*, *Ancylostoma* sp., *Trichuris trichiura*, *Loa loa*, *Mansonella perstans*, *Taenia saginata* and *Schistosoma haematobium*), and five protozoans (*Plasmodium*, *Entamoeba coli*, *Endolimax nana*, *Enteromonas* sp., and *Giardia* sp.). The mean number of parasite species per participant was 1.67 (±1.02). It was higher in participants living in rural areas (2.06 ± 0.78) than in urban areas (1.32 ± 1.08) (*p* = 1.0810 × 10^−15^). The mean was, respectively, 1.6 (±1.02) and 1.71 (±1.02) in females and males. There were more children infected with more than one parasite in the rural area (63.6%) than in the urban area (36.4%) (*X*^2^ = 94.5, *p* ˂ 2.210 × 10^−16^). Polyparasitism was more common in children aged between 11 and 17 than in those below 11 (58.6% vs. 41.4%). Gender had no influence (58.6% vs. 41.4%, *p* = 0.199). Double parasitism was more prevalent (75/99).

## 4. Discussion

The present study investigated the presence of asymptomatic malaria parasites and co-infections among schoolchildren in two Gabonese settlements. Although some studies have identified asymptomatic malaria parasites, data on their co-occurrence with other infections in Gabonese schoolchildren remain limited. 

This cross-sectional study identified various associations between asymptomatic malaria infection and other infections. Among co-infected individuals, 12.3% had IPIs, 4.7% had filarial infections, and 6.7% had *Schistosoma* infection. Previous studies in Gabon reported co-infection rates of 7% for IPIs [22], 0.2% for filaria [22], and 9% for *Schistosoma* [13], showing slight variations from our findings, possibly due to differences in target populations and sample sizes. Earlier studies primarily focused on febrile individuals, with such as 428 febrile vs. 88 afebrile [23]; 410 febrile vs. 60 afebrile [24]; and 793 febrile vs. 100 afebrile [25]. In contrast, our study exclusively screened about 470 afebrile children. Additionally, potential confounding factors such as socioeconomic, genetic, and nutritional should be considered in explaining these discrepancies. 

A high prevalence of asymptomatic malaria infection in children was observed at 69.7%, surpassing the 57.08% and 42.92% found in febrile and afebrile children from Lastourville [26], as well as the 52% in adults from Lambarene region [9]. Our findings exceed those from other African countries such as CAR, 35.2% [27], and Cote d’Ivoire, 50.3% [28]. These discrepancies may stem from limited access to diagnostics, treatment, and prevention during the COVID-19 pandemic, as well as seasonal and geographical variations among study populations [29,30]. High rates of asymptomatic malaria infection could impair local authorities‘ malaria elimination efforts, these individuals may serve as reservoirs for the malaria parasite and contribute to autochthonous transmission cycles [31,32]. Additionally, rural areas recorded higher malaria prevalence and parasitemia compared to urban areas, probably due to reduced access to control strategies [22], environmental factors, and high entomological inoculation rates [33]. This study confirms the heterogeneity of malaria burden and transmission intensity in Gabon [31]. Older children (11–17 years) were more likely to be infected with malarial parasites than younger children, which is consistent with previous reports [31]. Winskill et al. [34] suggest this pattern may relate to the use of insecticide-treated mosquito nets, as younger children tend to sleep under these protective nets more frequently [34]. However, other studies indicate that asymptomatic malaria prevalence decreases with age due to the development of protective immunity from cumulative exposure and increased knowledge of malaria prevention strategies [30].

In the present study, the prevalence of IPIs was found to be 19.1%, significantly lower than the 61.1% reported in different settlements of Gabon by M’bondoukwé et al. [22], and 49.0% in Lambarené by Adegnika et al. [35]. When compared to similar studies in other countries, the prevalence also falls below those reported in communities near Buea, Cameroon (47.2%) [36], Nigeria (24%) [37], Angola (44.2%) [38], and Mozambique (31.6%) [39]. Conversely, lower prevalence rates were observed in hospitals in Cameroon (11.9%) [40], Ghana (15%) [41], and Ethiopia (15.5%) [42]. Intestinal parasitic infections were more prevalent in male children than female children, regardless of the location and the parasite species, which is consistent with findings from other studies [41]. These discrepancies could be attributed to the behavior factors, males often play outside and have more contact with soil, while females are often involved in household chores. Among the isolated intestinal parasites, *Ascaris lumbricoides* was the predominant parasite in infected children, consistent with reports from other areas of Gabon [22] and elsewhere [37,38]. Co-infection with *Plasmodium* and IPI occurred at high levels compared to previous studies in Gabon [22], and elsewhere [43,44,45], but at low levels compared to studies in Ethiopia [46] and Cameroon [47]. However, no association was found between asymptomatic malaria and IPI, which is consistent with findings by Mantagila et al. [44].

The prevalence rates of microfilariae infection from *L. loa* and *M. perstans* were similar to those reported by M’bondoukwé et al. [22], but lower than previous studies in other areas of Gabon, where at least 15% of screened individuals had one or both worms [48,49]. Microfilariasis was more prevalent in rural areas than urban ones, in line with other studies in Gabon [22,50]. The current study found no association between filarial infection and malaria.

This study reports a significantly higher prevalence of *Schistosoma haematobium* compared to the 1.7% found by Mintsa-Nguema et al. [51] in 2018, but lower than the 26% in Libreville and Ekouk by Mintsa et al. [52] and, 30% near Lambaréné by Dejon-Agobe et al. [13]. These variations highlight the patchy distribution of schistosomiasis, influenced by human and ecological factors [53], as well as temperature and rainfall [54]. The co-occurrence of *Plasmodium* and *Schistosoma* indicates a significant association, suggesting a potential protective effect of one on the other [55]. This aligns with previous reports, although some studies suggest that *S. haematobium* may increase susceptibility to *P. falciparum* infection [56].

Anemia is a significant public health problem for schoolchildren in malaria-endemic areas, affecting physical growth, cognition, and academic performance. Its diverse causes in tropical areas include helminths, hemoglobinopathies, and malnutrition. To combat anemia, public health strategies such as long-lasting insecticide-treated nets (LLINs), artemisinin-based combination therapy (ACTs), indoor residual spraying and mass administration of antihelmintic drugs (MDAs) are implemented. A high prevalence of anemia (29%) was found among schoolchildren, with approximately 85% of anemic individuals carrying asymptomatic *Plasmodium* infection. This prevalence aligns with findings from Cameroon (30.8% [57]), Ethiopia (41.3% [30]), and Nigeria (34.4% [58]), but is lower than the >73.5% observed in febrile children in Gabon [59]. Anemia was significantly more common in children under five (*p* = 0.000), with no notable gender difference. Young children are particularly vulnerable to anemia, worsened by bacterial infections, malaria, and intestinal parasites. Our findings indicate a significant association between anemia and malaria, supporting previous studies that highlight the severe impact of malaria and related infections on anemia in schoolchildren [44,60,61,62].

The findings from this study have implications for managing malaria-related morbidities in the two involved settlements, and possibly in other regions with similar conditions. However, there are limitations: parasitological microscopy cannot detect asymptomatic infections at the submicroscopic level, and relying on single slides for STHs may yield biased results due to variability in STH egg excretion, especially in children with low-intensity parasitemia. Additionally, the study design does not support causality assessments between asymptomatic malaria and other causes of anemia. Furthermore, our study did not consider the historical timeline of prior malaria infections, which could provide valuable insights into latent parasitemia and its interaction with other infections. Without information on past infection history, it is difficult to correlate latent malaria infections with other co-infections or health outcomes over time. Future studies could benefit from incorporating such longitudinal data to better understand the dynamics between asymptomatic malaria and co-infections in these populations.

## 5. Conclusions

Asymptomatic malaria infection is highly prevalent, along with intestinal parasitic infections, filaria and *Schistosoma*, they remain major public health challenges among school-aged children in Franceville and Larstourville. This study is the first to document the prevalence of these co-infections and also highlight their correlation with anemia in children in these regions of Gabon. The findings emphasize the need for ongoing surveillance, improved diagnostic techniques, and strengthened prevention efforts to mitigate the silent burden of malaria and co-infections in vulnerable populations.

## Figures and Tables

**Table 1 tropicalmed-10-00011-t001:** Characteristic of the study population according to the study area.

	Urban n (%)	Rural n (%)	Total
Gender (N = 469)			
Male	122 (49.2)	100 (45.2)	222 (47.3)
Female	126 (50.8)	121 (54.7)	247 (52.7)
Total	248 (52.9)	221 (47.1)	469
Sex ratio	0.96	0.82	0.9
Age group (N = 460)			
3–10	138 (55.4)	111 (44.6)	249 (54.1)
11–17	101 (47.9)	110 (52.1)	211 (45.9)
Total	239 (51.9)	221 (48.1)	460
Parameters			
Mean temperature ± SD (°C)	37.0 ± 0.4	36.7 ± 0.8	36.9 ± 0.7
Mean age ± SD (year)	9.7 ± 3.2	10.3 ± 3.1	10.04 ± 3.19
Haemoglobin (g/dL)	11.8 ± 1.3	11.2 ± 1.3	11.66 ± 1.4
WBC (×10^3^/µL)	7.3 ± 2.1	9.3 ± 9.5	8.08 ±5.92
RBC (×10^6^/µL)	4.42 ± 0.5	2.49 ± 4.9	4.01 ± 2.48
Platelet (×10^3^/µL)	294.2 ± 107.1	1983.7 ± 1154.7	838.95 ± 1029.79

**Table 2 tropicalmed-10-00011-t002:** Prevalence of malaria, IPI, filaria and *Schistosoma* by location, age groups and gender.

		Malaria	IPI	Filaria	*Schistosoma*
Location					
	Urban n (%)	119 (47.8)	51 (24.7)	2 (0.8)	NA
	Rural n (%)	208 (94.1)	26 (13.3)	22 (10.1)	32 (14.5)
	*p*-value	<2.2 × 10^−16^	0.0055	1.522 × 10^−05^	-
Age groups					
	3–10	163 (65.5)	38 (18.2)	6 (2.4)	11 (12.2)
	11–17	162 (76.8)	38 (20.3)	18 (8.7)	21 (19.4)
	*p*-value	0.0089	0.680	0.0055	0.101
Gender					
	Male	158 (71.2)	47 (24.5)	10 (4.5)	17 (17.9)
	Female	168 (68.0)	30 (14.4)	14 (5.7)	15 (12.8)
	*p*-value	0.5218	0.0145	0.7273	0.4046
Total	n (%)	327 (69.6)	77 (19.2)	24 (5.1)	32 (15.1)

NA: Not Applicable.

**Table 3 tropicalmed-10-00011-t003:** Risk factors associated with malaria in logistic models (simple and multiple).

	Univariate				Multivariate			
	95% CI				95% CI			
	OR			*p*-Value	OR			*p*-Value
		Lower	Upper			Lower	Upper	
Gender (male)	1.161	0.783	1.726	0.459	1.386	0.858	2.254	0.185
Age	1.124	1.054	1.200	<0.001	1.037	0.909	1.179	0.587
Weight	1.020	1.002	1.039	0.037	1.023	0.987	1.064	0.222
Temperature	1.009	0.991	NA	0.654	1.565	1.081	2.259	0.015
City (LTV)	17.627	9.882	33.996	<0.001	23.649	12.256	49.767	<0.001

Abbreviations: OR, odds ratio; 95% CI, 95% confidence interval, NA, Not Applicable.

**Table 4 tropicalmed-10-00011-t004:** Risk factors associated with IPI in logistic models (simple and multiple).

	Univariate				Multivariate			
	95% CI				95% CI			
	OR			*p*-Value	OR			*p*-Value
		Lower	Upper			Lower	Upper	
Gender (male)	1.934	1.170	3.239	0.011	1.732	1.028	2.950	0.040
Age	1.057	0.976	1.145	0.175	1.183	1.024	1.371	0.023
Weight	1.006	0.983	1.028	0.626	0.967	0.926	1.009	0.127
Temperature	0.988		1.010	0.782	0.845	0.588	0.989	0.379
City (LTV)	0.468	0.275	0.780	0.004	0.377	0.206	0.672	0.001

Abbreviations: OR, odds ratio; 95% CI, 95% confidence interval.

**Table 5 tropicalmed-10-00011-t005:** Risk factors associated with filaria in logistic models (simple and multiple).

	Univariate				Multivariate			
	95% CI				95% CI			
	OR			*p*-Value	OR			*p*-Value
		Lower	Upper			Lower	Upper	
Gender (male)	0.789	0.334	1.802	0.577	0.829	0.327	2.043	0.685
Age	1.328	1.149	1.559	<0.001	1.384	1.063	1.838	0.020
Weight	1.040	1.007	1.073	0.014	0.983	0.915	1.050	0.615
Temperature	1.137	1.044	2.292	0.699	1.826	1.038	3.869	0.098
City (LTV)	13.806	4.003	86.835	<0.001	14.016	3.786	91.433	0.001

Abbreviations: OR, odds ratio; 95% CI, 95% confidence interval.

**Table 6 tropicalmed-10-00011-t006:** Association of asymptomatic malaria infection with different infections (IPI, filaria and *Schistosoma*).

Variables	Univariate analysis		
	OR	95% CI	*p*-Value
IPI	1.02	0.669–1.57	0.927
Filaria	1.99	0.797–6.04	0.174
*Schistosoma haematobium*	0.0867	0.0440–0.157	2.70 × 10^−14^

Abbreviations: OR, odds ratio; 95% CI, 95% confidence interval.

**Table 7 tropicalmed-10-00011-t007:** Correlations between different infections and anemia.

	IPI	Filaria	Malaria	Anemia Status
IPI	1			
Filaria	−0.009	1		
Malaria	0.040	0.110	1	
Anemia status	−0.015	0.030	0.270	1

## Data Availability

The datasets used and/or analyzed during the current study are available from the corresponding author upon reasonable request.

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
