# Peer review of "Asymptomatic Malaria Infection and Hidden Parasitic Burden in Gabonese Schoolchildren: Unveiling Silent Co-Infections in Rural and Urban Settings"

_tropicalmed, 2024, doi:10.3390/tropicalmed10010011_

Round 1

Reviewer 1 Report

Comments and Suggestions for Authors

Minor language errors (capitalization, no plural, typos). In several places there is only a reference to literature without providing the author's name.

Line 61:  one verb is enough

Line 109: urine samples.

Line 110: was collected

Line 113: urine was used for ...

Line 125: blood samples.

Line 140: name is missing here [19].

Line 142: performed on

Line 151: microscopic examination

Line 158: urban

Line 159: characteristics

Line 167: Characteristics of the study population, according to the study area.

Line 184: The rates of intestinal ...

Line 196: Loa loa

Line 199: participants

Line 204: malaria, filariasis

Line 208: stood

Line 230: living / residing

Line 240: gender, age

Line 241: city, age

Line 266: participants

Line 285: found

Line 289: ruled

Line 293: showed a higher prevalence than …    studies performed …

Line 295: COVID-19 pandemic

Line 304: counterparts

Line 305: name is missing here [34].

Line 315: name is missing here [22],   name is missing here [35].

Line 321: which is consistent with ...

Line 328: those reported …  name is missing here [22],  studies in …

Line 329: least

Line 348: infections,

Line 353: give

Line 354: intensity parasitemia.

Line 396: participated

Reviewer 2 Report

Comments and Suggestions for Authors

The manuscript entitled, "Asymptomatic Malaria and Hidden Parasitic Burden in Gabonese Schoolchildren: Unveiling Silent Co-infections in Rural and Urban Settings," offers valuable insights into the prevalence and co-infection status of asymptomatic malaria cases. I commend the authors on their work and have a few minor suggestions to further refine the manuscript:

  1. Latent Infections and Timeline Correlation: Given that many children might have experienced prior malaria infections, it would be valuable to explore whether latent parasitemia was considered. Including an analysis of the historical timeline of malaria infections in the study participants, and potentially correlating these with other diseases, would strengthen the discussion.
  2. Typographical Review: A typographical error was noted in line 208, which should be corrected.
  3. Clarification of Coinfection Observations: The sentence between lines 224-226, stating, "Notably, all children infected with microfilaria parasites were also coinfected with malaria parasites with no cases of combined microfilaria infection, i.e., they either harbored Plasmodium sp. plus Loa loa or Plasmodium sp. plus Mansonella perstans," may benefit from rephrasing for clarity. I suggest rewording to indicate that all children with microfilaria infections were simultaneously infected with malaria parasites, but no children harbored concurrent infections of both Loa loa and Mansonella perstans alongside Plasmodium sp.
  4. Comparison of Disease Prevalence with and without Co-infection: It would be beneficial to include a comparative analysis of the prevalence of other diseases in children with and without malaria co-infections. Additionally, discussing whether co-infection with malaria could serve as a confounding variable in the prevalence of other diseases would enhance the findings.

Overall, with these minor revisions, I believe the manuscript holds significant merit and is worthy of publication.

Thank you for the opportunity to review this work.

Reviewer 3 Report

Comments and Suggestions for Authors

This manuscript is valuable for the prevention and control of Plasmodium and other parasites in school-age children in Africa,But there are some shortcomings.

1. The whole inspection method lacks duplication, and the results will be biased.

2. The table is written in a standardized format, some can be added to the graph display.

3.Ancylosyomasp, and Giardiasp  , et al Please check whether such terms are correct?

4.Collating statistics.

5. Write the results clearly and succinctly.

6.Rewrite the conclusion.

7.Full text needs to sulk.

Comments on the Quality of English Language

/

Reviewer 4 Report

Comments and Suggestions for Authors

The manuscript requires extensive standardization, as explained below

 Title – Asymptomatic malaria is not a happy conjugation of terms. Malaria is a disease, while asymptomatic means the absence is symptoms. This reviewer would suggest: Asymptomatic Plasmodium sp. infection… or Asymptomatic malaria infection…

 Please standardize capitalization (uppercase/lowercase) in the title

 Line 27 – malaria is not an infection but a disease – please rewrite to read as: … infection with Plasmodium sp.

 Remark: if there is more than one species of Plasmodium in Gabon, use Plasmodium spp. Alternatively, use Plasmodium (without sp. or spp.) – change accordingly throughout the manuscript

 Line 31 – who provided informed consent: the children or their parents/legal guardians? Adapt

 Line 34 – merthiolate-iodine concentration (MIC) – change accordingly in the main text

 Line 35 – A total of 470 school-aged / IPI (instead of IPIs)

 Line 36 – prevalence is not a rate but a proportion – replace with prevalence values

 Line 41 – present p values

 Line 45 – replace rates with prevalences

 Line 46 – Lastourville or LTV? Please standardize

 Keywords – display in alphabetical order

 Line 50 – replace significant with important

 Remark: please differentiate diseases from subclinical infections. A parasitic disease has always an underlying infection, but an infection by itself may be asymptomatic/subclinical. So, Plasmodium infection and malaria are not necessarily the same thing.

 Line 54 – revise: Plasmodium infection (WITH malaria) is a leading cause of death

 Line 55 – define WHO

 Line 58 – intestinal parasitic infections (IPI)

 Line 59 – helminths (STH), protozoa, or both

 Line 64 – WHO has defined “asymptomatic malaria infection” (and not asymptomatic) – please adapt accordingly throughout the manuscript

 Line 76 – Plasmodium sp. or Plasmodium spp.? How many species are the authors talking about?

 Line 82 – write P. falciparum at its second and subsequent uses – change accordingly regarding the names of the other parasites

 Line 102 – 3 months

 Line 105 – The study WAS a …

 Line 113 – write out Schistosoma hematobium – first use

 Line 116 – thick blood smears (TBS)

 Line 117 – what is 20% Giemsa?

 Line 120 – how many microliters do 100 oil immersion fields represent?

 Line 121, etc. – microfilariae

 Line 135 – 1 minute

 Line 1412 – microhaematuria

 Line 157 – 470 (instead of words) – change accordingly

 Lien 159 – Table 1

 Line 169 – Three hundred and twenty-seven (n = 327)

 Line 170, etc. – these percentages should be compared using statistical tests

 Line 172 – write Plasmodium in italics

 Lines 180-190 – these percentages should be compared using statistical tests

 Line 188 – trichiura / Giardia spp.

 Line 192 – Filarial INFECTIONS WERE more common in THE rural area than in THE urban area

 Remark: percentages should be compared in order to determine if the differences are statistically significant

 Line 206 – microfilariae – change accordingly

 Line 210 – adapt: Among the 57 children with asymptomatic malaria INFECTION and IPIs, the most

 Remark – write “spp.” when there is more than one possible species

 Standardize text in the Results and Discussion sections as recommended above

 Line 353 – it is not necessary to explain the meaning of abbreviation STH once again

 References – write Latin names in italics; use lowercase in titles, as much as possible; check the title format of ref. 33

Round 2

Reviewer 3 Report

Comments and Suggestions for Authors

The revised draft has been greatly improved, but there are still some deficiencies:

1. The formatting of subheadings and some paragraphs does not meet the requirements,such as 115,127-128Merthiolate-iodine concentration (MIc ),139,et al.

2. Statistical method description is missing.

3.Collating statistics.

Comments on the Quality of English Language

/

Author Response

  1. The formatting of subheadings and some paragraphs does not meet the requirements, such as 115,127-128Merthiolate-iodine concentration (MIc ),139,et al. Response: corrected as suggested
  2.  Statistical method description is missing; Response: Thank you for the feedback. A statistical section was provided (L156-170)
  3. Collating statistics. Responses: As stated before, keeping separate tables allows for a more organized and accessible presentation of our results. Unless the reviewer has a specific demand, please be more specific. 

Reviewer 4 Report

Comments and Suggestions for Authors

The authors have addressed all of my comments and accepted all of my suggestions.

Author Response

The reviewer made no comment that required an answer from us, and we want to thank him for his help. 

Round 3

Reviewer 3 Report

Comments and Suggestions for Authors

The author has made great improvement, some problems are basically solved, and it is suggested to publish.